# 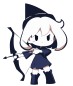 Artemis: Towards Referential Understanding in Complex Videos

**Jihao Qiu**[1]* **Yuan Zhang**[1]* **Xi Tang**[1]* **Lingxi Xie** **Tianren Ma**[1]

**Pengyu Yan**[2] **David Doermann**[2] **Qixiang Ye**[1] **Yunjie Tian**[1,2]†

[1]University of Chinese Academy of Sciences  [2]University at Buffalo

{qiujiahao19, zhangyuan192, tangxi19, matianren18, tianyunjie19}@mails.ucas.ac.cn
pyan4@buffalo.edu  doermann@buffalo.edu  198808xc@gmail.com  qxye@ucas.ac.cn

## Abstract

Videos carry rich visual information including object description, action, interaction, *etc.*, but the existing multimodal large language models (MLLMs) fell short in referential understanding scenarios such as video-based referring. In this paper, we present **Artemis**, an MLLM that pushes video-based referential understanding to a finer level. Given a video, Artemis receives a natural-language question with a bounding box in any video frame and describes the referred target in the entire video. The key to achieving this goal lies in extracting compact, target-specific video features, where we set a solid baseline by tracking and selecting spatiotemporal features from the video. We train Artemis on the newly established VideoRef45K dataset with 45K video-QA pairs and design a computationally efficient, three-stage training procedure. Results are promising both quantitatively and qualitatively. Additionally, we show that Artemis can be integrated with video grounding and text summarization tools to understand more complex scenarios. Code and data are available at `https://github.com/qiujihao19/Artemis`.

## 1 Introduction

The past year has witnessed rapid progress of multimodal large language models (MLLMs) [33, 68, 40], offering abundant abilities of open-world image understanding with language-based dialogues. In comparison, there are fewer studies on training MLLMs for video understanding, albeit videos are much more informative than still images. Existing video-based MLLMs [29, 61, 37, 31, 38] mostly focus on superficial dialogues in which the video is encoded holistically, inevitably lacking the ability to understand fine-level video contents, *e.g.*, describing a user-specific target in the video.

We are considering a new task called video-based referential understanding to compensate for the limitation. Specifically, we are interested in complex videos that span 20–30 seconds and the target performs multiple actions during this period. Given a video, the MLLM tries to answer a question like 'What is the target <region> doing in this video?' where <region> refers to a bounding box in any video frame. We argue that the task is not only challenging as it requires feature extraction, tracking, summarization, *etc.*, but also important because it lays the foundation of finer-level video understanding. However, as shown in Figure 1, existing MLLMs often fell short in

---
* Equal contribution. † Corresponding Author.

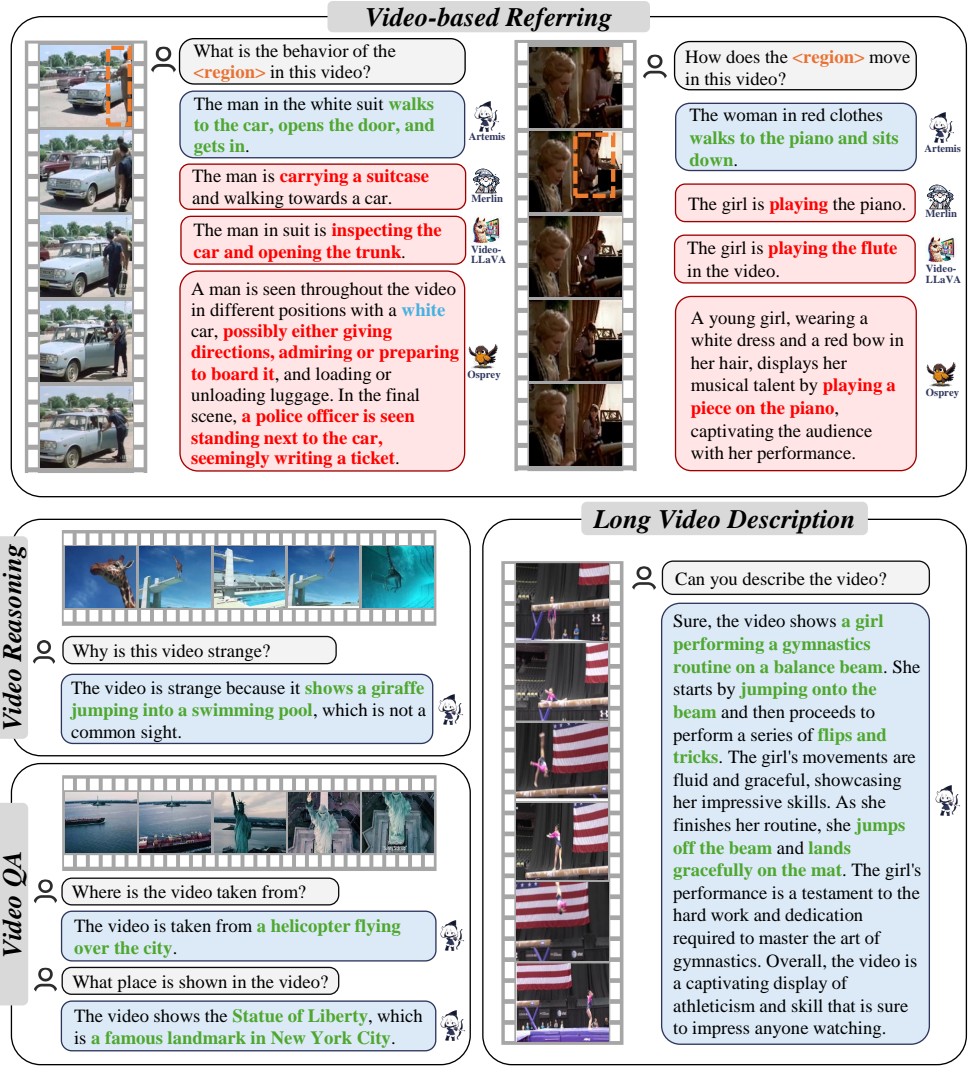

Figure 1: Artemis' ability in video-based dialogue. Notably, Artemis excels particularly in video-based referring, outperforming the existing MLLMs including Merlin [57] and Video-LLaVA [31] lacking comprehensiveness and Osprey [59] suffering hallucination.

this seemingly easy task because they were mostly trained for image-based referential understanding; as a result, they can only perceive the action in a single moment rather than that in an entire video[2].

This paper presents **Artemis**[3] as a solid baseline for the above task. Artemis follows the generic design of modern MLLMs (*i.e.*, visual instruction tuning [33]), but encounters a challenge in finding *sparse*, target-related information from *dense* video data. A preliminary study shows that feeding raw video features into the MLLM results in computational inefficiency and training instability. To extract target-specific video features, we propose a simple yet effective solution that involves (i) tracking the target over time and (ii) selecting informative features from a long list of regions-of-interest (RoIs). The compactness of features makes it easier to train the MLLM. We design a three-stage training schedule where the MLLM gradually learns video-text alignment from coarse to fine. This efficient design requires only 28 hours (3 hours for the final stage) on $8\times$ NVIDIA-A800 GPUs.

To train and evaluate Artemis, we organize 7 existing video understanding datasets into the **Video-Ref45K** benchmark comprising 45K video question-answer pairs. To our knowledge, this is the first

---

[2]One can also use an image-based model for frame-wise referring and call an LLM for text summarization, but, as shown in Appendix C, this method often fails to understand the intrinsic logic in complex videos.

[3]The name refers to Artemis' ability to track prey and select pivotal hunting moments.

benchmark with box-level prompts and answers spanning complex videos. Experiments show the promising results of Artemis in a wide range of quantitative metrics including the BERT score, BLEU, *etc.*. Qualitatively, Artemis also shows a clear advantage in the comprehensiveness of description meanwhile avoiding hallucination (see Figure 1 for examples). Beyond the ability of video-based referring, Artemis serves as an important building block for complex video understanding, where we integrate Artemis with off-the-shelf video grounding and text summarization tools for interactive video-based dialogue and long video understanding, respectively. We expect our work to shed light on upgrading MLLMs for fine-level and interactive video understanding.

## 2   Related Work

**Large language models (LLMs) and multimodal LLMs (MLLMs).** LLMs [15, 5, 13, 45, 12, 64, 49, 60, 11] have opened a new era of AI, demonstrating the potential to deal with various language-based understanding and generation tasks. To unleash the power of LLMs for visual understanding, the computer vision community has been working on aligning language and vision data in the same feature space [41]. There are mainly two lines of research, where the *internal* adaptation methods [1] integrated cross-attention within an LLM for visual-language alignment, and the *external* adaptation methods [27, 14, 33] trained extra modules for this purpose. As a result, the vision foundation models, especially vision transformers [17, 35, 41, 48, 47, 66, 25], have been upgraded into MLLMs [33, 46, 63, 26] which gain the ability of language-guided visual understanding.

**MLLMs for referring and grounding.** MLLMs can be integrated with instance-level visual understanding tasks, allowing the models to (i) respond to questions targeted at specific regions of the image and (ii) identify regions corresponding to the contents in the dialogue – these functions are referred to as visual referring [63, 7] and grounding [40, 34], respectively. There are two main ways to integrate these functions into MLLMs, differing from each other in how the positional information is processed. The *explicit* methods [40, 52] introduced extra tokens to encode positions, while the *implicit* methods [9, 51, 54] used natural language to represent positions. Recently, there are also efforts [46] that used LLMs to call external vision modules for more flexible instance-level understanding quests.

**Video-based MLLMs.** Compared to the large corpus of image-based MLLMs, there are fewer video-based MLLMs for at least two reasons. First, there are fewer paired video-text data, especially for instance-level video understanding. Second, the higher dimensionality of video data poses a greater challenge to efficiently encode videos into visual features and find useful features to answer the questions. Existing efforts include VideoChat [29], Video-ChatGPT [37], Video-LLaMA [61], Video-LLaVA [31], LanguageBind [67], Valley [36], *etc.*; most of them followed the paradigm of image-based MLLMs and some of them [37] proposed a more efficient video feature. Recently, there have been some preliminary studies for instance-level video understanding, *e.g.*, LEGO [30] studied moment retrieval with the assistance of LLMs, and PG-Video-LLaVA [38] performed video grounding by employing off-the-shelf tracking and grounding modules. Merlin [57] studied video-based referring, but it was built upon three manually specified frames as visual input, incurring extra burden for users and also limiting the model's ability to understand long and complex videos. *This paper aims to address the above two challenges, for which we set up a new formulation, establish a new benchmark named VideoRef45K, and present a solid baseline named Artemis.*

## 3   Artemis: A Baseline for Video-based Referential Understanding

### 3.1   Problem Formulation and Data Preparation

A video can be represented in the raw form of $\mathbf{V} \in \mathbb{R}^{T \times W \times H \times C}$, where $T$, $W$, $H$, and $C$ stand for the number of frames, width, height, and the number of channels, respectively. In the task of video-based referential understanding (*a.k.a.* video-based referring), the model receives a question in the form of 'What is the <region> doing in this video?', where the concrete class of the referred object (like man or dog) is not provided, and the <region> is supplemented by a bounding box $\mathbf{B} = (t; x_1, y_1, x_2, y_2)$ in a frame $t \in \{1, 2, \ldots, T\}$. The expected output is a sentence describing the target's action in the full video as detailed as possible (see Figure 1 for examples). Note that the proposed task requires a stronger ability beyond image-based referring and video captioning,

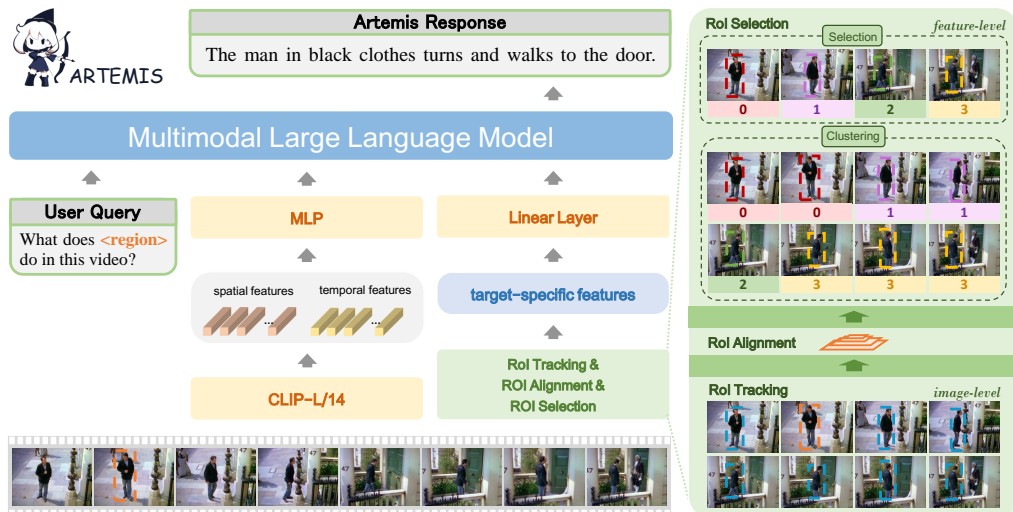

Figure 2: **Left**: the overall framework of Artemis, where an MLLM receives a text prompt together with spatial, temporal, and target-specific video features, and produces the answer. **Right**: the RoI tracking and selection mechanism to generate target-specific features. We use different IDs to show the clustering result. *This figure is best viewed in color.*

mainly in the coverage and granularity of visual understanding. Specifically, the model is expected to produce complex action descriptions for different `target <region>` specified.

We collect video data for referential understanding from 7 datasets, including HC-STVG [44], VID-Sentence [10], A2D Sentences [20], LaSOT [18], MeViS [16], GOT10K [24], and MGIT [23]. In total, there are 45K video QA pairs. We perform dataset-specific operations, including re-tracking (for HC-STVG and A2D-Sentences), clip cropping (for LaSOT and MGIT), and caption summary (for GOT10K), to convert them into the required form. Please refer to Appendix A for further details.

### 3.2 Overall Framework and Visual Features

The overall framework of Artemis, as illustrated in Figure 2, follows the pipeline of visual instruction tuning [33, 68] where a multimodal large language model (MLLM) receives video features with a text prompt and produces the desired output. We denote the function as $\mathbf{T}_{\text{out}} = f(\mathbf{F}_{\mathbf{V}}, \mathbf{T}_{\text{in}})$, where $\mathbf{T}_{\text{in}}$ and $\mathbf{T}_{\text{out}}$ are input and output texts (in tokens) and $\mathbf{F}_{\mathbf{V}}$ is the set of features extracted from $\mathbf{V}$.

Compared to image-based referring, a clear difficulty of video-based referring arises from the high dimensionality of video data. Specifically, if we define $\mathbf{F}_{\mathbf{V}}$ as the set of dense video features (*e.g.*, using a pre-trained visual encoder such as the CLIP ViT-L model [41] to extract frame-wise visual features for $\mathbf{V}$), the features often contain highly redundant information due to the similarity of neighboring frames. This brings two-fold drawbacks: (i) extra complexity for the MLLM to deal with these vision tokens, and (ii) extra difficulty for the MLLM to locate useful information, which leads to a slower convergence. To overcome this issue, we decrease the input feature dimensionality by using various slices to replace $\mathbf{F}_{\mathbf{V}}$, where each slice captures important yet complementary properties of the input video. Throughout this paper, we investigate three slices: the spatial, temporal, and target-specific video features.

**Spatial and temporal features.** The extraction of spatial and temporal video features follows the design of Video-ChatGPT [37]. Given a video clip $V \in \mathbb{R}^{T \times W \times H \times C}$, we use the CLIP ViT-L/14 visual encoder to cast it into frame-wise features, denoted as $F_{\text{frame}} \in \mathbb{R}^{T \times W' \times H' \times D}$, where the number of frames remains unchanged, $W' = W/s$ and $H' = H/s$ are the down-sampled resolution (*e.g.*, $s = 14$ for ViT-L/14) of the visual features, and $D$ is the feature dimensionality.) Then, these features are fed into average pooling along the $T$ axis (into the spatial features $\mathbf{F}_{\mathbf{V}}^{\text{S}} \in \mathbb{R}^{W' \times H' \times D}$) and along the $W' \times H'$ plane (into the temporal features $\mathbf{F}_{\mathbf{V}}^{\text{T}} \in \mathbb{R}^{T \times D}$), respectively.

**Target-specific features.** $\mathbf{F}_{\mathbf{V}}^{\mathrm{S}}$ and $\mathbf{F}_{\mathbf{V}}^{\mathrm{T}}$ have focused on the spatial and temporal features but ignored the referred `target` which may move or change during the video. To offer a compromise feature that captures spatiotemporal features, we propose an RoI (region-of-interest) tracking and selection mechanism (detailed in Section 3.3) and obtain a list of RoIs (represented as bounding boxes) $\mathcal{B} = (\mathbf{B}_1, \ldots, \mathbf{B}_M)$, where $M$ is the number of RoIs that are recognized by the algorithm to be important for referential understanding. We use the RoIAlign method [21] to extract visual features from each RoI, producing a set of target-specific features, $\mathcal{F}_{\mathbf{V}}^{\mathrm{R}} = (\mathbf{F}_{\mathbf{V}, \mathbf{B}_1}^{\mathrm{R}}, \ldots, \mathbf{F}_{\mathbf{V}, \mathbf{B}_M}^{\mathrm{R}})$.

**Instruction fine-tuning.** When the video features are ready, we feed them with the text tokens into Artemis. The MLLM follows instruction fine-tuning through three steps, gradually acquiring the ability of video-based referring. The details are described in Section 3.4.

## 3.3 RoI Tracking and Selection

Our goal is to extract compact features for video-based referring. The key lies in two factors, (i) completeness – locating the referred target in every video frame, and (ii) avoiding redundancy – not preserving too many features in the frames with similar semantics. We propose a simple solution upon RoI tracking and selection. As we shall see later, it offers a solid baseline for future work.

**Step 1: RoI tracking.** We apply HQTrack [69], an off-the-shelf tracking algorithm, to localize the RoI in each input frame. The pre-trained tracking model is not fine-tuned in the training phase. Given a RoI (a bounding box) in any video frame, the tracking algorithm outputs either a bounding box or nothing (*e.g.*, if the target is occluded) in each of the remaining frames. This step outputs a raw list of RoIs denoted as $\mathcal{B}' = (\mathbf{B}'_1, \ldots, \mathbf{B}'_{M'})$ where $M'$ can be close to the number of frames.

**Step 2: RoI selection.** Feeding all tracked frames into the MLLM often incurs computational inefficiency and extra difficulties in model training. To avoid this, we select a subset from $\mathcal{B}'$ containing $M < M'$ RoIs, with the goal being to preserve diverse visual features using a limited number of RoIs. In practice, we pre-defined the target number, $M$, and adopt the K-means algorithm to form $M$ clusters from the original set of $M'$ RoIs. The final RoI list, $\mathcal{B}$, consists of a randomly chosen RoI from each cluster.

**Discussions.** Finding representative RoIs belongs to a generic topic of feature selection. On one hand, one can set a simple baseline by performing random or uniform sampling from the original set $\mathcal{B}'$. On the other hand, the information theory offers a general principle, *i.e.*, maximize the diversity of RoIs throughout the selection procedure. As demonstrated in Section 4.1, random and uniform sampling algorithms frequently fail to capture semantic changes throughout complex videos. By contrast, the simple K-means clustering used in Artemis significantly increases the diversity (see Appendix D), ensuring representative video features. We conjecture that the effectiveness of feature selection is related to the quality of video features; with stronger video foundation models, more sophisticated feature selection algorithms can make a larger difference. We leave this topic to future research.

## 3.4 Model Architecture and Training

The MLLM is built upon Vicuna-7B v1.5 [11], an open-sourced LLM[4]. We use CLIP ViT-L/14 [41] to extract visual features. To feed these 1024-dimensional visual tokens into the LLM, we use a learnable, two-layer MLP (1024-4096-4096) to project the visual features into the 4096-dimensional language space. We always use the auto-regressive framework to train the MLLM.

The training procedure of Artemis comprises three steps, (1) video-text pre-training, (2) video-based instruction tuning, and (3) video-based referring. The first two stages are similar to Video-LLaVA [31] but different training data are used. We set a unified template,



`User:` `<video-tokens> <instruction>` `Assistant:`



guiding the model to output the desired answer. Here, `<video-tokens>` contains the spatial and temporal video features ($\mathbf{F}_{\mathbf{V}}^{\mathrm{S}}$ and $\mathbf{F}_{\mathbf{V}}^{\mathrm{T}}$, projected by MLP), and `<instruction>` contains the language tokens of the task description (see below).

In the first stage, `<instruction>` has the form of '`Write a terse but informative summary of the following video clip.`' and the model outputs the overall description of the video.

---

[4]A stronger LLM (*e.g.*, with a larger number of parameters) brings marginal improvement, because at the current stage, video understanding does not rely on strong language modeling abilities.

The training data includes image-text and video-text pairs, using images as still videos. We use a subset of 558K LAION-CCSBU image-text pairs with BLIP [28] captions, sourced from CC3M [42] and refined by LLaVA [33]. Additionally, we use the 702K video-text pairs provided by Video-LLaVA [31], derived from the 703K pairs constructed by Valley [36] using WebVid [3]. Only the MLP is trained (from scratch) in this stage, initializing the alignment of vision and language. The training elapses one epoch with a learning rate of $1 \times 10^{-3}$, taking about 5 hours on 8×A800 GPUs.

In the second stage, `<instruction>` contains specific task descriptions like 'Where is the person in the image?' and 'What is the person doing in the video?', and the model follows the instruction to produce the answer. The training data comprises the 665K image-text instruction dataset from LLaVA-1.5 [33] and the 100K video-text instruction set from Video-ChatGPT [37]. Both the LLM and MLP are fine-tuned in this stage. The training elapses one epoch with a learning rate of $2 \times 10^{-5}$, taking about 20 hours on 8×A800 GPUs.

In the third stage, we use the curated VideoRef45K dataset to endow the model with the ability of video-based referring. The template is modified as follows,

  User: `<video-tokens> <refer-instruction> <track-instruction>` Assistant:

Here, `<refer-instruction>` is formulated as 'What is the `<region>` doing during this video?' where the `<region>` token is replaced by the visual features extracted from the bounding box in the specified input frame, and `<track-instruction>` contains additional information, 'This is the tracking list: `<region>`, ..., `<region>`' where the `<region>` tokens are the target-specific features $(\mathbf{F}^{R}_{\mathbf{V},\mathbf{B}_1}, \ldots, \mathbf{F}^{R}_{\mathbf{V},\mathbf{B}_M}$, projected by a Linear) extracted from the selected RoIs[5], and the number of `<region>` token is $M$. In this stage, we fine-tune the LLM (with LoRA [22]), MLP and the RoI Align module. The training procedure elapses 3 epochs with a learning rate of $4 \times 10^{-5}$, taking about 3 hours on 8×A800 GPUs.

## 4 Experiments

### 4.1 Artemis Is a Strong Baseline for Video-based Referential Understanding

**Setting and metrics.** We evaluate the ability of Artemis in video-based referring on the test set of HC-STVG [44]. The video and text data are pre-processed using the same method as in the training set. The test procedure uses the same instruction as in the third training stage and applies HQTrack [69] to localize the RoIs in video frames. We use the standard evaluation metrics including BERTScore [65], BLEU@4 [39], METEOR [4], ROUGE_L [32], CIDEr [50], and SPICE [2].

Table 1: A comparison of video-based referring metrics on the HC-STVG test set. [†]: We use 5 key frames while using 8 frames leads to worse results.

| Method | BERT Score | BLEU@4 | METEOR | ROUGE_L | CIDEr | SPICE |
|---|---|---|---|---|---|---|
| Osprey [59] | 0.8698 | 0.7 | 12.0 | 18.0 | 1.2 | 15.6 |
| Ferret-13B [56] | 0.8632 | 0.5 | 10.2 | 17.0 | 1.2 | 11.2 |
| Shikra-7B [9] | 0.8742 | 1.3 | 11.5 | 19.3 | 3.1 | 13.6 |
| Video-ChatGPT [37] | 0.8718 | 1.3 | 10.1 | 20.2 | 5.5 | 11.7 |
| Video-LLaVA [31] | 0.8639 | 1.7 | 9.8 | 20.8 | 2.6 | 9.1 |
| Merlin [57][†] | 0.8829 | 3.3 | 11.3 | 26.0 | 10.5 | 20.1 |
| Artemis (Ours) | 0.9135 | 15.5 | 18.0 | 40.8 | 53.2 | 25.4 |

**Adapting existing MLLMs for video-based referring.** Due to the limited availability of research for video-based referring, we compare our model to a few recent MLLMs trained for image-based or multi-frame based referring[6]. The image-based referring models include Osprey [59], Ferret [56],

---

[5]During the training phase, we randomly select a frame with an annotated region bounding box as the input and employ the tracking module to locate the bounding box of the referred object in the sampled frames. During inference, we track the given region to generate the tracking list.

[6]The comparison against these methods is not totally fair because they have not been trained for video-based referring. We mainly use the comparison to claim that video-based referring is important and challenging, yet image-based MLLMs cannot do it well.

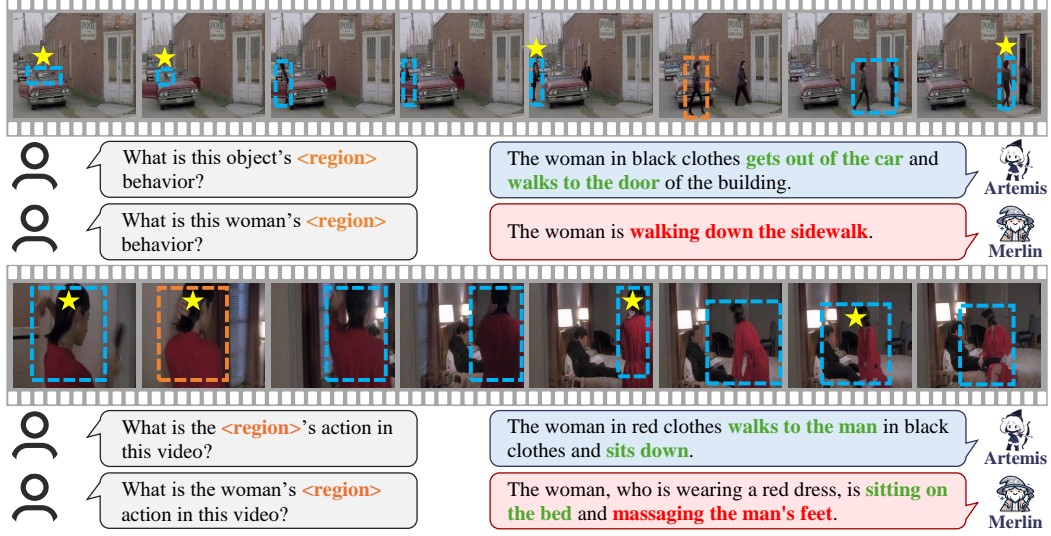

Figure 3: Artemis and Merlin for video-based referring. Note that Merlin needs the semantic class of `<region>` to be provided while Artemis does not. In each case, the orange rectangle indicates the input `<region>`, blue rectangles are the tracked RoIs, and yellow stars label the selected RoIs. Red and green texts indicate incorrect and correct answers, respectively. *This figure is best viewed in color.*

and Shikra [9]. For each video, we extract 5 key frames with RoIs produced by HQTrack and ask the trained model "`What is the target <region> doing?`" in the way the models are familiar with. Finally, we use GPT-3.5-Turbo to summarize the 5 answers into the overall description of the target. The multi-frame based reference model is Merlin [57] which receives 5 key video frames and RoIs and produces the overall description. The selection of key frames is consistent with Artemis. To compare with MLLMs that are trained for video understanding, such as Video-ChatGPT [37] and Video-LLaVA [31], we follow [43] to draw a red rectangle to mark the referred object in each key frame of the video. Then, we feed the rendered video to the models and ask the question "What is the target indicated by the red rectangle doing?".

**Quantitative results, and necessity of native video-based referring.** The numbers are summarized in Table 1. Artemis outperforms other MLLMs in each single evaluation metric. Note that the advantage is significant for some metrics, *e.g.*, BLEU4. Please refer to Figure 1 for representative examples. In Figure 9 (see Appendix C), we show the behavior of the methodology using a standalone LLM (*e.g.*, GPT-3.5-Turbo) upon image-based referring outputs. The image-based models tend to describe individual moments rather than an entire video; based on these inputs, the LLM cannot realize video descriptions and is sometimes confused to hallucinate what never happens in the video. The comparison validates the necessity of training a **native** model (*i.e.*, directly on the instruction data for video-based referring) like what Artemis has done. Equipping with such a fundamental ability of video understanding at a finer level, Artemis can perform even more complex video understanding tasks, as shown in Section 4.2.

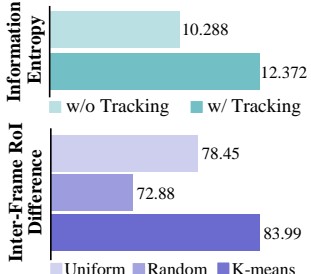

Figure 4: RoI manipulation increases the informativeness and diversity of RoIs. See Appendix D for details.

**Qualitative results.** We display several representative examples of video-based referring in Figure 3. The output of Artemis is comprehensive (especially compared to other MLLMs, see Figure 1), often containing fine-grained actions of the target. This mainly concerns the compact video features extracted by the RoI tracking and selection algorithm that extracts key features for understanding.

**Ablative studies for target-specific video features.** The key to extracting compact target-specific features lies in RoI tracking and selection. To validate this, we ablate two key parameters: the strategy of RoI selection and the number of preserved RoIs. In Table 2, we define a baseline for region of interest. For each frame of a video containing an object of interest, we enclose the object's location

with a red rectangle and encode the video using Video-ChatGPT. The system is then queried with "What is the object in the red rectangle doing in this video?". As illustrated in Table 2, K-means clustering emerges as a simple yet effective approach for RoI selection, whereas random or uniform sampling fails to consistently capture representative RoIs. To validate the effectiveness of RoI features in representing the object, we replace the <region> features with visual features (the [CLS] token from CLIP-ViT-L/14) in key frames, a variation we refer to as "w/<track-instruction>". The performance decline compared to Artemis indicates that RoI features are of higher quality, as whole-frame features are more susceptible to background noise. Additionally, Table 3 demonstrates the importance of using multiple RoIs for understanding the full video content. While retaining more keyframes slows down both training and inference, it also introduces redundant information, leading to a slight drop in performance metrics. Empirically, using 4 RoIs provides the optimal balance on the HC-STVG test set, although increasing the number of RoIs may be beneficial for more complex videos.

From the information theory, we show that RoI tracking and selection improve informativeness (in terms of entropy) and diversity (in terms of frame-level difference) of the target-specific features in Figure 4. As shown in Figure 5, RoI tracking and selection gradually improve the comprehensiveness of the referring results.

Table 2: Ablation on different RoI selection methods. Results are reported on HC-STVG.

| Method | BLEU@4 | METEOR | ROUGE_L | CIDEr | SPICE |
|---|---|---|---|---|---|
| baseline | 11.2 | 16.3 | 34.9 | 23.8 | 21.4 |
| w/o | 13.9 | 16.9 | 39.1 | 43.7 | 23.2 |
| w/<track-instruction> | 13.9 | 16.9 | 38.2 | 42.1 | 23.1 |
| Uniformly | 14.2 | 17.2 | 39.4 | 44.5 | 23.6 |
| Randomly | 14.3 | 17.1 | 40.0 | 46.5 | 24.2 |
| Clustering | 14.6 | 17.4 | 40.2 | 47.2 | 23.9 |

Table 3: Ablation on the number of selected RoIs. Results are reported on HC-STVG.

| # RoI | Bert Score | BLEU@4 | METEOR | ROUGE_L | CIDEr | SPICE |
|---|---|---|---|---|---|---|
| 1 | 0.9114 | 13.9 | 16.9 | 39.0 | 43.3 | 23.6 |
| 2 | 0.9113 | 14.6 | 17.5 | 39.5 | 43.5 | 23.7 |
| 4 | 0.9125 | 14.3 | 17.1 | 40.0 | 46.5 | 24.2 |
| 6 | 0.9122 | 14.3 | 16.9 | 39.5 | 46.2 | 23.9 |
| 8 | 0.9110 | 14.0 | 17.0 | 39.0 | 43.1 | 23.6 |

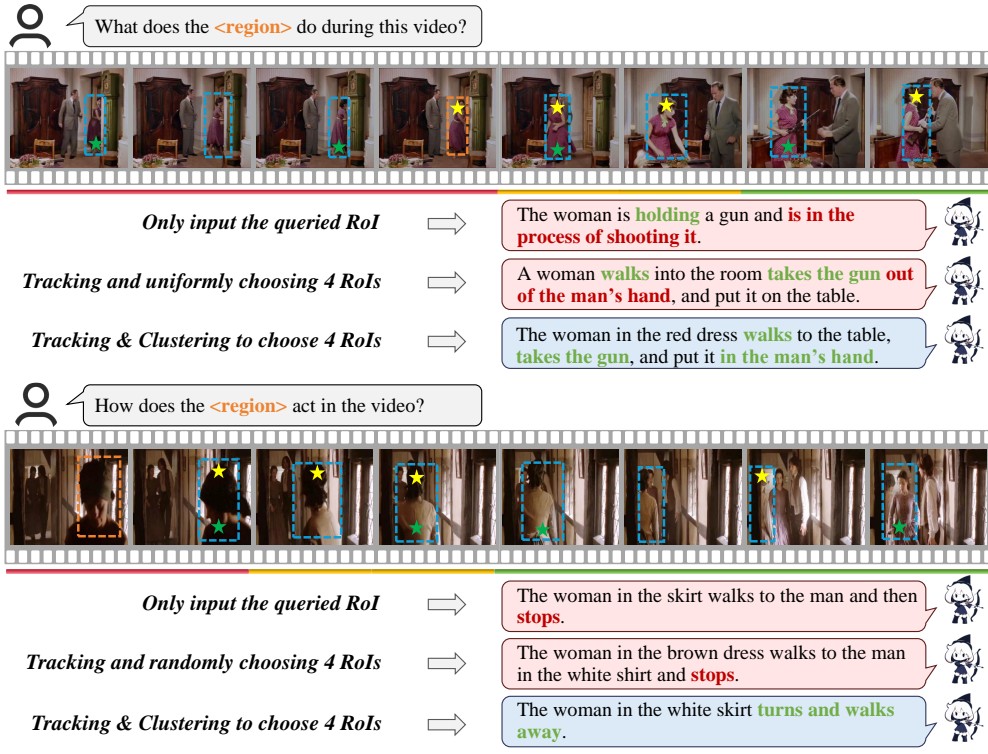

Figure 5: How RoI tracking and selection gradually improves the quality of video-based referring. In each example, the orange rectangle indicates the input <region>, blue rectangles are the tracked RoIs, and green and yellow stars label the uniformly sampled and K-means selected RoIs, respectively. Red and green texts highlight the incorrect and correct outputs. *This figure is best viewed in color.*

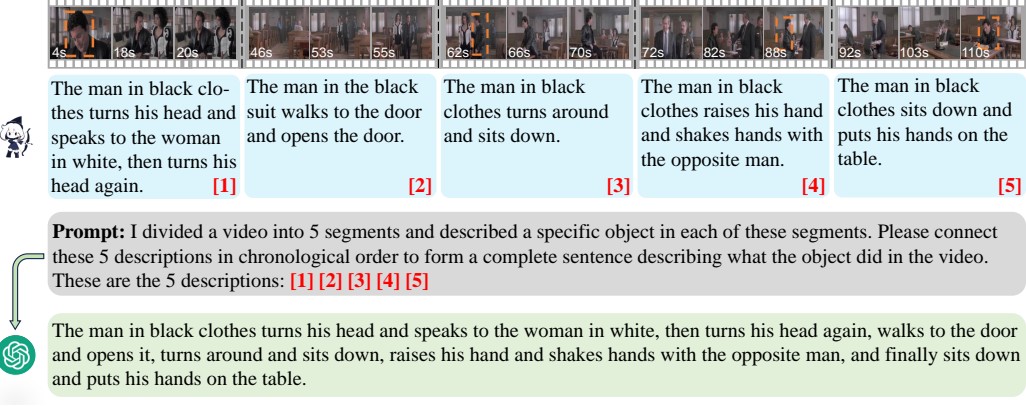

**Prompt:** I divided a video into 5 segments and described a specific object in each of these segments. Please connect these 5 descriptions in chronological order to form a complete sentence describing what the object did in the video. These are the 5 descriptions: [1] [2] [3] [4] [5]

The man in black clothes turns his head and speaks to the woman in white, then turns his head again, walks to the door and opens it, turns around and sits down, raises his hand and shakes hands with the opposite man, and finally sits down and puts his hands on the table.

Figure 7: Example of long video understanding. We apply Artemis to output descriptions for segmented video clips and integrate them using an LLM (GPT-3.5-Turbo in this example).

## 4.2 Artemis Is a Building Block for Complex Video Understanding

With a strong ability of video-based referring, Artemis serves as a building block that strengthens the existing video-based MLLMs in complex video understanding.

**Multi-round video understanding with grounding.** Multi-round dialogues, especially answering logically related chain-of-questions [46], is an important yet challenging topic for MLLMs. In Figure 6 and Figure 14 in Appendix E, we show that Artemis's referential understanding ability can be combined with image-based grounding models (*e.g.*, GroundingDINO [34]) to answer multi-round chain-of-questions, where the entities mentioned in the video-based referring result is located and fed into the next round of video-based referring quest, allowing for more complex interactions.

**Long video understanding with text summarization.** Target-centric understanding of long videos is a major challenge for existing video-based MLLMs. The difficulty mainly lies in extracting compact video features (to feed into the MLLM) and tracking the target throughout the long video. We offer a simple solution that first segments the video into shorter clips, applies Artemis for understanding these clips, and applies an off-the-shelf LLM (*e.g.*, GPT-3.5-Turbo) for summarization. As shown in Figure 7 and Figure 12 in Appendix E, the final output offers a comprehensive understanding. To our knowledge, this function was not achieved by existing MLLMs.

Figure 6: An example of multi-round, video-based referring by integrating Artemis with GroundingDINO [34].

**Video question answering.** Lastly, we show that Artemis can perform general video question answering. We test the trained model on the Video-ChatGPT test set [37] and the other three benchmarks (*i.e.*, MSVD-QA [8], MSRVTT-QA [53], and ActivityNet-QA [6, 58]) where their training sets was not used to train Artemis. Results are summarized in Tables 4. Artemis shows competitive performance among a few recent MLLMs. These results inspire us that (i) an MLLM trained for finer-level video understanding can seamlessly transfer to coarser-level tasks, and (ii) extracting compact video features also benefits video question answering.

## 5 Conclusion

This paper proposes a challenging setting for video-based referring and establishes an effective MLLM named Artemis. Compared to existing methods, Artemis can understand human intention from simpler inputs (a text prompt and a single-frame bounding box) and comprehensively describe the target's action in a complex video. At the core of Artemis is an RoI tracking and selection

Table 4: **Left**: Video QA on Video-ChatGPT. Metrics: correctness (CN), detail orientation (DO), contextual understanding (CU), temporal understanding (TU), consistency (CC). **Right:** Zero-shot video QA on MSVD-QA, MSRVTT-QA, and ActivityNet-QA. Metrics: accuracy (Acc.), score (Sc.).

| Method | CN | DO | CU | TU | CC |
|---|---|---|---|---|---|
| Video-Chat [29] | 2.23 | 2.50 | 2.53 | 1.94 | 2.24 |
| LLaMA-Adapter [62, 19] | 2.03 | 2.32 | 2.30 | 1.98 | 2.15 |
| Video-LLaMA [61] | 1.96 | 2.18 | 2.16 | 1.82 | 1.79 |
| Video-ChatGPT [37] | 2.40 | 2.52 | 2.62 | 1.98 | 2.37 |
| Valley-v3 [36] | 2.43 | 2.13 | 2.86 | 2.04 | 2.45 |
| Artemis (Ours) | 2.69 | 2.55 | 3.04 | 2.24 | 2.70 |

| Method | MSVD Acc. | MSVD Sc. | MSRVTT Acc. | MSRVTT Sc. | ActivityNet Acc. | ActivityNet Sc. |
|---|---|---|---|---|---|---|
| FrozenBiLM [55] | 32.2 | - | 16.8 | - | 24.7 | - |
| Video-Chat [29] | 56.3 | 2.8 | 45.0 | 2.5 | 26.5 | 2.2 |
| LLaMA-Adapter [62, 19] | 54.9 | 3.1 | 43.8 | 2.7 | 34.2 | 2.7 |
| Video-LLaMA [61] | 51.6 | 2.5 | 29.6 | 1.8 | 12.4 | 1.1 |
| Video-ChatGPT [37] | 64.9 | 3.3 | 49.3 | 2.8 | 35.2 | 2.7 |
| Valley-v3 [36] | 60.5 | 3.3 | 51.1 | 2.9 | 45.1 | 3.2 |
| Artemis (Ours) | 72.1 | 3.9 | 56.7 | 3.2 | 39.3 | 2.9 |

mechanism to extract compact video features. Artemis shows advantages in video-based referring in VideoRef45K and transfers the ability to general video understanding, including being integrated with other modules for more complex tasks. We hope that Artemis can serve as a solid baseline to facilitate the research in fine-level video understanding.

**Limitations.** First, Artemis relies on a tracking algorithm to generate the RoIs; however, the tracking algorithm may produce inaccurate results and can confuse Artemis– see an example in Figure 13 (top) in Appendix E. Second, Artemis also suffers from the issues of general video-based understanding, such as the spatial-temporal aliasing problem, which can affect the model's ability to describe the visual content accurately – see an example in Figure 13 (bottom) where Artemis accurately predicts the movement of the target but reverses the temporal order.

## 6 Acknowledgments

This work was supported by National Natural Science Foundation of China (NSFC) under Grant No.62225208 and CAS Project for Young Scientists in Basic Research under Grant No.YSBR-117.

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

# A  Data Curation

We organized 7 existing video datasets and performed a careful data curation procedure, resulting in the VideoRef45K benchmark, which comprises 45K video question-answer pairs. The 7 datasets include HC-STVG [44], VID-Sentence [10], A2D Sentences [20], LaSOT [18], MeViS [16], GOT10K [24], and MGIT [23].

HC-STVG is a movie clip dataset that provides tracking sequences and textual annotations describing the person's actions during a certain period. We use the training portion, which contains approximately 10K video clips as our training data. The validation portion, containing 3,400 video clips, evaluates Artemis's ability. The original tracking sequences in HC-STVG are of poor quality, so we use the off-the-shelf tracking model HQTrack [69] to regenerate the tracking sequences and remove some low-quality bounding boxes. To prevent tracking target deviation caused by cross-frame tracking, we select the first and middle frames with ground truth bounding boxes annotated by the HC-STVG dataset as the referring frames for HQTrack to generate tracking lists for the whole video. We then compare these two generated tracking lists with the HC-STVG annotations and exclude the frames with low IoU between the generated and annotated bounding boxes from the tracking lists.

A2D Sentences provides tracking sequences and captions for different objects, but the tracking sequences are only 3 frames long. To address this, we use HQTrack to regenerate the sequences and obtain longer tracking frames, extending them to 20 frames.

LaSOT provides a caption for an object along with its tracking sequence. However, LaSOT videos are usually long, and the captions for the entire video are generic. To address this, we extract three segments of 10 seconds each from the entire video for our training data.

GOT-10K is a tracking dataset that provides tracking sequences of objects and their categories, actions, and adverbs describing the actions. We concatenate these elements to describe the object's action in the video, *e.g.*, "bear is slowly walking."

MGIT videos are typically long, with annotations indicating the object's actions at different time intervals. We extract these segments as our training data.

For MeViS and VID-Sentence, we did not perform any special processing. We converted the mask annotations of MeViS into bounding boxes.

Figure 8 shows some examples of VideoRef45K.

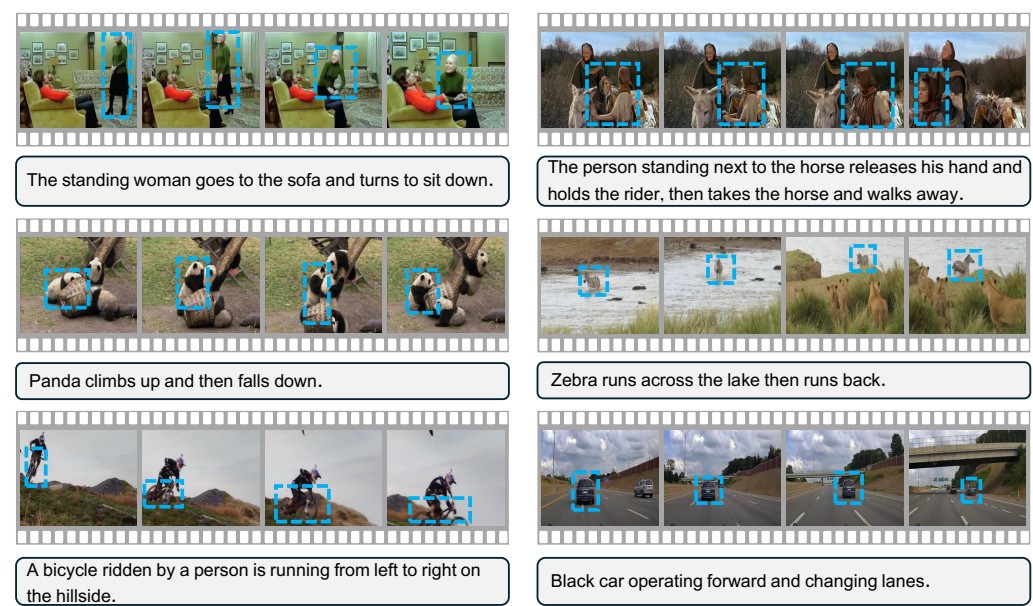

Figure 8: Some examples of VideoRef45K.

The total dataset composition is summarized in Table 5.

Table 5: Curated datasets for Video-based Referring.

| Dataset | Video Clips | Q&A Pairs |
|---|---|---|
| HC-STVG [44] | 10105 | 10105 |
| MeViS [16] | 1644 | 4489 |
| A2D Sentences [20] | 3017 | 5359 |
| LaSOT [18] | 2640 | 7920 |
| VID Sentence [10] | 4045 | 7654 |
| GOT-10K [24] | 9090 | 9090 |
| MGIT [23] | 105 | 614 |
| **VideoRef45K** | **30646** | **45231** |

Through the above processing steps, we obtained the VideoRef45K dataset. This dataset includes captions describing the actions of objects in videos, along with their corresponding tracking sequences. To utilize VideoRef45K, we created a template of questions to prompt the language model (LLM) to answer what the referred object did in a video. The template consists of '<refer-instruction>' and '<track-instruction>'. The '<refer-instruction>' is formulated as '`What is the <region> doing during this video?`', and we utilized GPT-3.5-Turbo to generate the refer instruction template. The '<track-instruction>' contains additional tracking lists to help the LLM perceive the referred object, such as '`This is the region's tracking list: <region> ... <region>`'. We created several '<track-instruction>' options as the track instruction template. During training, we randomly sampled a '<refer-instruction>' from the refer instruction template and a '<track-instruction>' from the track instruction template. The '<refer-instruction>' and the '<track-instruction>' are then concatenated as '<refer-instruction><track-instruction>' to formulate the text prompt.

## B Implementation Details

We report the detailed training hyper-parameters of Artemis in Table 6.

Table 6: Training hyper-parameters of Artemis.

| Configuration | Pre-training | Instruction Tuning | Referring Instruction Tuning |
|---|---|---|---|
| ViT init. | CLIP-L/14 | CLIP-L/14 | CLIP-L/14 |
| LLM init. | Vicuna-7B-v1.5 | Vicuna-7B-v1.5 | Artemis-Finetune |
| Projection init. | random | Artemis-Pretrain | Artemis-Finetune |
| Image resolution | 224 | 224 | 224 |
| Video feature length | 356 | 356 | 356 |
| LLM sequence length | 2048 | 2048 | 2048 |
| Optimizer | AdamW | AdamW | AdamW |
| Peak learning rate | $1 \times 10^{-3}$ | $2 \times 10^{-5}$ | $4 \times 10^{-5}$ |
| Minimum learning rate | 0 | 0 | $4 \times 10^{-5}$ |
| Learning rate schedule | cosine decay | cosine decay | constant |
| Weight decay | 0 | 0 | 0 |
| LoRA rank | None | None | 16 |
| Number input trackbox | None | None | 8 |
| Number choosen bbox | None | None | 4 |
| Training steps | 4927 | 5979 | 142 |
| Global batch size | 256 | 128 | 48 |
| Numerical precision | bfloat16 | bfloat16 | float16 |

## C Image-based MLLMs for Video-based Referring

Existing multimodal language models (MLLMs) for video understanding rarely possess video-based referring capabilities. To address this limitation, we leverage image-based MLLMs (such as Osprey [59] and Ferret [56]) to perform image referring on 5 keyframes within the video independently. Subsequently, we integrate these outputs using GPT-3.5-Turbo to achieve video-based referring, as

depicted in Figure 9. However, these models struggle to perceive the action and behavior of the given region of interest (RoI) as effectively as Artemis.

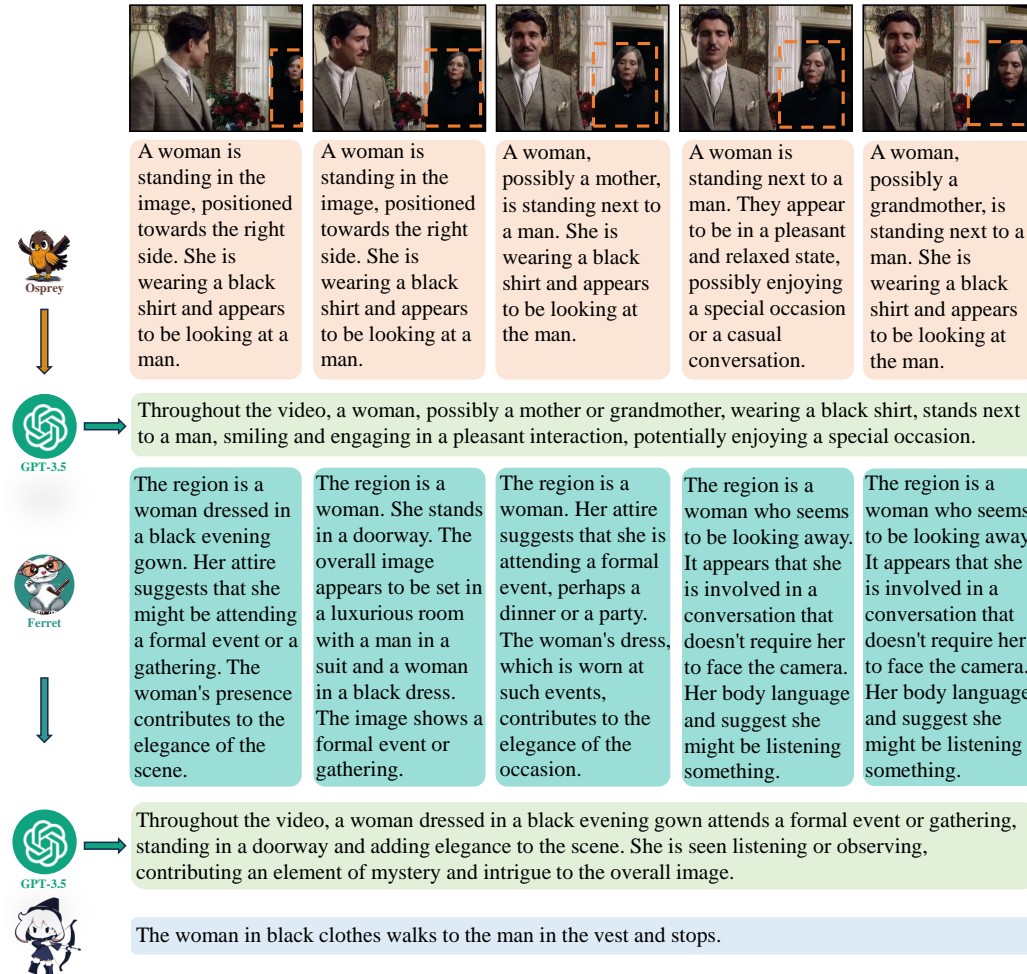

Figure 9: A comparison of video-based referring between image-based MLLMs and Artemis. GPT-3.5-Turbo is used to integrate the 5 independent outputs from the image-based MLLMs.

## D   How RoI Tracking and Selection Improve the Results

To better demonstrate the effectiveness of our method, we conducted the following experiments. Firstly, we computed the attention values between different RoI tokens and temporal tokens. We observed that the tracking tokens added through tracking compensated for the weak perception aspect of the initial video tokens, as illustrated in Figure 10. Subsequently, as depicted in Figure 4 (top), we calculated the information entropy before and after adding the tracking list. Upon adding the tracking list, the overall amount of RoI information fed into the MLLM increased by 20.3%. Additionally, we computed the inter-frame differences of the boxes chosen from the tracking list. As shown in Figure 4 (bottom), the K-means clustering method selects RoIs with greater differences than random and average selection. This enables the MLLM to better perceive the action changes of RoIs throughout the entire video.

## E   More Qualitative Examples

Figure 11 shows more examples of Artemis's outputs of video-based referring. Figure 12 shows more examples of target-centric video understanding with text summarization. Figure 13 shows

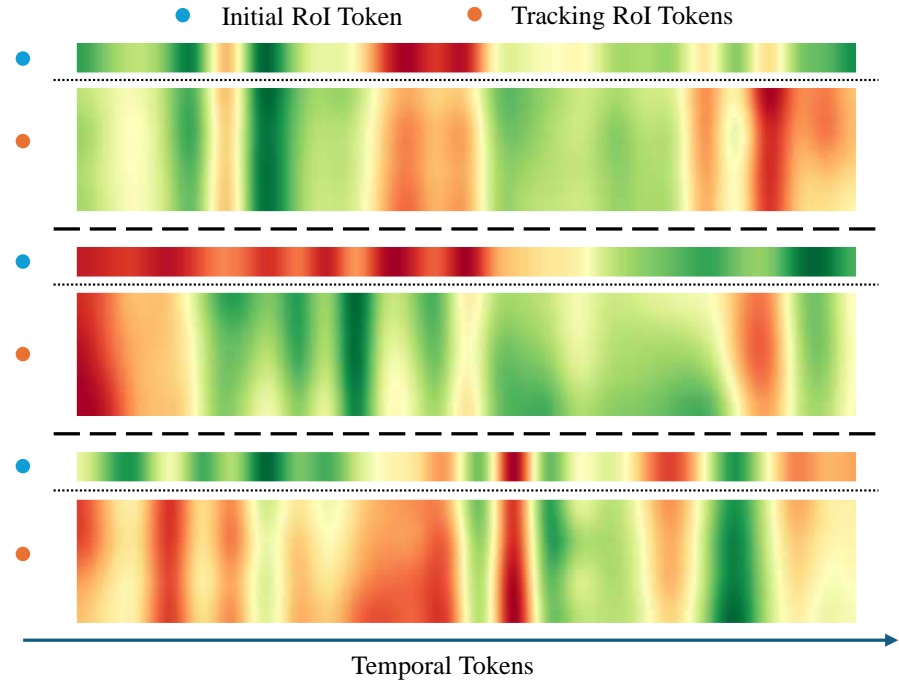

Figure 10: Attention map between RoI tokens and temporal tokens.

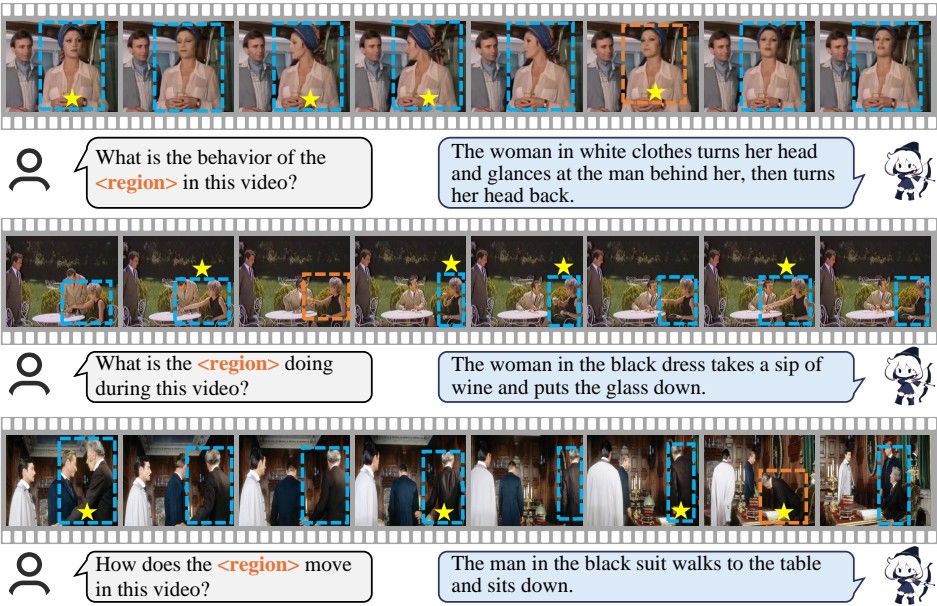

Figure 11: Examples of video-based referring generated by Artemis.

Artemis's failure cases, revealing the limitation of Artemis. Figure 14 shows mores example of combining Artemis with off-the-shelf grounding model, *e.g.* GroundingDINO to answer multi-round chain-of-questions conveniently.

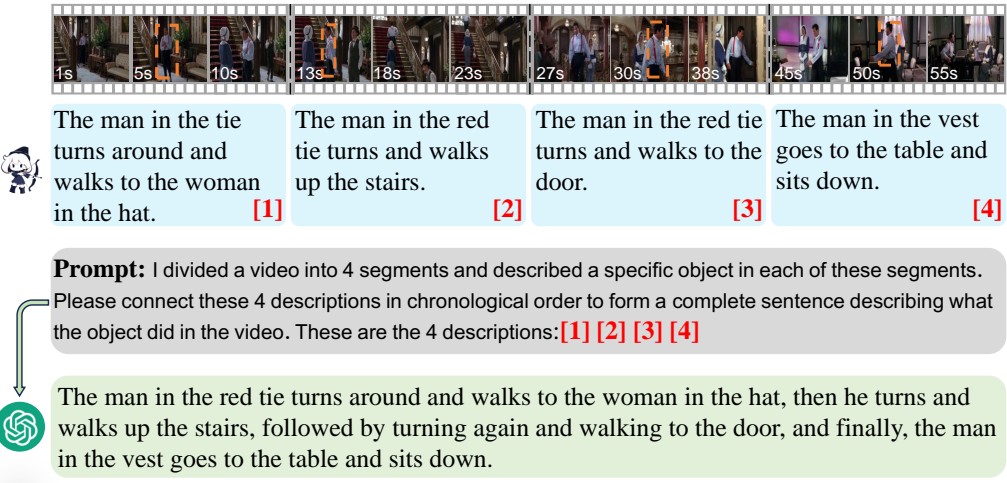

Figure 12: The example of long video understanding generated by Artemis.

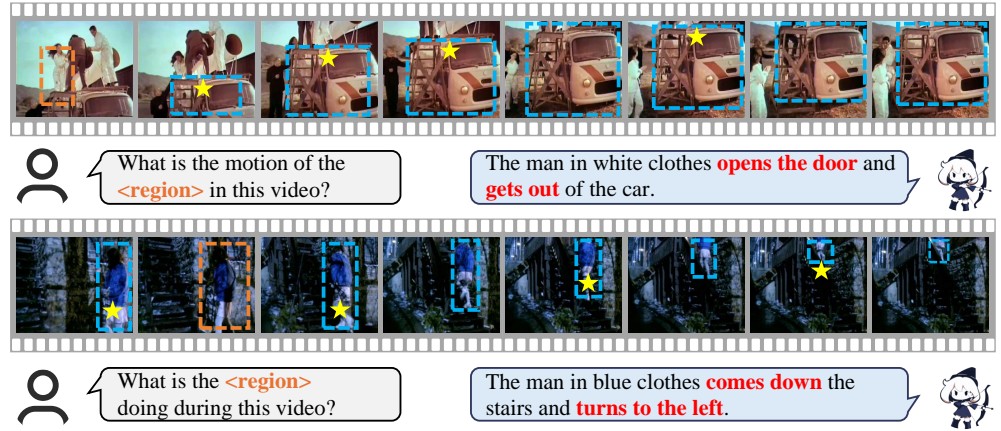

Figure 13: Failure cases of Artemis. **Top**: The tracking module generates inaccurate RoI list, misleading Artemis's understanding. **Bottom**: Spatial-temporal aliasing in video hinders Artemis to perceive objects.

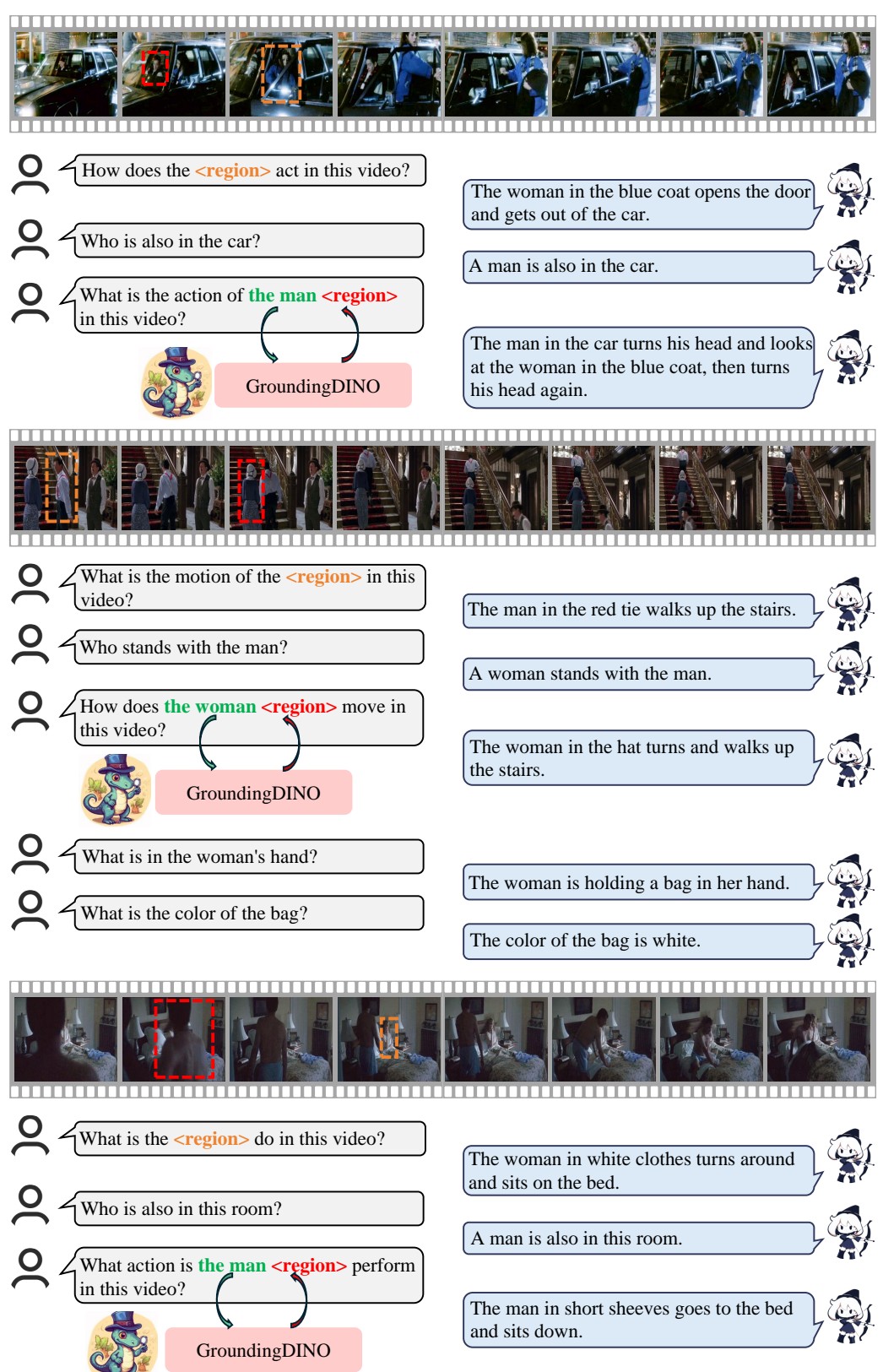

Figure 14: Examples of multi-round video understanding with grounding generated by Artemis.

