# OpenReview forum: "Artemis:  Towards Referential Understanding in Complex Videos"
_NeurIPS.cc/2024/Conference — NeurIPS 2024 poster_

### Official Review · Reviewer_xzsp · 2024-07-04

**Soundness:** 2
**Presentation:** 4
**Contribution:** 4
**Rating:** 7
**Confidence:** 5

**Summary:**

This paper introduces Artemis, a video-language model for video-based referential understanding. It can describe a target referred by a bounding box in a video. A referential video dataset named VideoRef45K is collected to train the model. Artemis is evaluated on HC-STVG benchmark and outperforms baselines adapted from image-based referring models. Besides referential understanding, Artemis can also perform general video question answering, and serve as a component in multi-round and long-form video understanding.

**Strengths:**

1. A video-based referential understanding dataset, VideoRef45K, is established. It facilitates the development of the area, providing referential pretraining data.
2. The design of target-specific feature branch in the model architecture is well-motivated.
3. Although no video-based referring model exists, the paper adapts image-based referring models to video as baselines. The adaption method is quite reasonable.
4. Artemis outperforms the adapted image-based baselines significantly on HC-STVG.
5. Artemis can still perform general video question answering and achieve better performance after training on the video referring task. This demonstrates that video referring can boost the reasoning capability of video-language models.
6. Combined with existing video-language models, Artemis can perform multi-round video understanding with grounding and long-form video understanding.

**Weaknesses:**

1. The RoI selection step of Artemis clusters the object bounding boxes from different frames. However, the clustering algorithm only considers the bounding box coordinates but does not take the visual content in the bounding boxes into account. In some cases, the bounding box of an object may remain unchanged for a long time but the object state keeps changing, e.g., a person standing at a certain location performs a series of actions. Clustering these frames together can compressing them into one would lose valuable information.
2. The proposed Artemis is built based on video-language models Video-LLaVA and Video-ChatGPT. However, these video-language models are not used as baselines in the experiments. Although they are not originally developed for video referring, there is a simple approach to adapt them for the task. As suggested by [1], directly drawing a circle or a rectangle on an image can help VLMs focus on the indicated object. Therefore, one can adapt the video-language models for video referring by drawing the object bounding boxes on the video frames and ask the model "What is the target indicated by the red rectangle doing?". Artemis should be compared with this simple baseline to demonstrate the effectiveness of the RoI feature branch in its model architecture.

[1] Shtedritski et al. What does CLIP know about a red circle? Visual prompt engineering for VLMs. ICCV 2023.

**Questions:**

1. Does the target-specific features in \<region\> tokens include the positional information of the bounding boxes?

**Limitations:**

Limitations are discussed in the paper.

---

> ### Author Rebuttal · Authors · 2024-08-07
>
> Thank you for recognizing the contributions of our work. We deeply appreciate your constructive comments and have provided point-to-point responses below. We hope our responses address all your concerns, and further comments are welcomed.
>
> **Q1:** *The RoI selection step of Artemis clusters the object bounding boxes from different frames. However, the clustering algorithm only considers the bounding box coordinates but does not take the visual content in the bounding boxes into account. In some cases, the bounding box of an object may remain unchanged for a long time but the object state keeps changing, e.g., a person standing at a certain location performs a series of actions. Clustering these frames together can compressing them into one would lose valuable information.*
>
>
> **A1**: Thanks for the question. We **did** take visual contents within the bounding box into consideration. Specifically, we computed a token for each RoI which contained the visual features extracted from the bounding box (from the same visual encoder, *i.e.* a pre-trained CLIP ViT-L/14 model) and used the token for clustering. We will add the implementation details to the final paper.
>
>
> **Q2:** *The proposed Artemis is built based on video-language models Video-LLaVA and Video-ChatGPT. However, these video-language models are not used as baselines in the experiments. Although they are not originally developed for video referring, there is a simple approach to adapt them for the task. As suggested by [1], directly drawing a circle or a rectangle on an image can help VLMs focus on the indicated object. Therefore, one can adapt the video-language models for video referring by drawing the object bounding boxes on the video frames and ask the model "What is the target indicated by the red rectangle doing?". Artemis should be compared with this simple baseline to demonstrate the effectiveness of the RoI feature branch in its model architecture.*
> > [1] Shtedritski et al. What does CLIP know about a red circle? Visual prompt engineering for VLMs. ICCV 2023.
>
>
> **A2**: Thanks for the suggestion! During the rebuttal, we evaluated Video-LLaVA and Video-ChatGPT for video referring using the method you proposed. Specifically, we followed [1] to draw a red rectangle to mark the referred object in each key frame of the video -- please note that tracking is required to mark the object in most frames. Then, we fed the rendered video to the models and asked the question "What is the target indicated by the red rectangle doing?" Results are summarized in the following table. One can see that, even with the help of an offline tracking algorithm, both Video-LLaVA and Video-ChatGPT report significantly lower scores compared to Artemis. This validates the effectiveness of Artemis' design. We will add these contents to the final paper.
>
> | Method | BLEU@4 | METEOR | ROUGE_L | CIDEr | SPICE |
> | :---: | :---: |:---: |:---: |:---: |:---: |
> | Video-ChatGPT | 1.3 | 10.1 | 20.2 | 5.5 | 11.7 |
> | Video-LLaVA | 1.7 | 9.8 | 20.8 | 2.6 | 9.1 |
> | Artemis (Ours) | 15.5 | 18.0 | 40.8 | 53.2 | 25.4 |
>
> **Q3:** *Does the target-specific features in <region> tokens include the positional information of the bounding boxes?*
>
> **A3**: Yes. We used positional encoding to incorporate the coordinates of the bounding boxes. Therefore, the <region> token contains the positional information.

---

> > ### Comment · Reviewer_xzsp · 2024-08-11
> >
> > I read all the reviews and the authors' responses to them. The responses to my comments are satisfactory and I highly appreciate the additional experiments. Therefore, I raised my rating to 7 (Accept).

---

> ### Author Response · Authors · 2024-08-11
> **Thanks**
>
> We are delighted that our response addressed your question. We appreciate your support for our work.

---

### Official Review · Reviewer_CDQu · 2024-07-12

**Soundness:** 3
**Presentation:** 3
**Contribution:** 3
**Rating:** 7
**Confidence:** 4

**Summary:**

This paper introduces Artemis as a robust solution for the video-based referential understanding task. This task involves analyzing complex videos, each spanning 20–30 seconds, where the target performs multiple actions. Given a video, the Multimodal Large Language Model (MLLM) attempts to answer questions such as "What is the target <region> doing in this video?" with <region> referring to a bounding box in any video frame. Artemis follows the general design principles of modern MLLMs, such as visual instruction tuning. To extract target-specific video features, Artemis employs a straightforward yet effective approach involving (i) tracking the target over time and (ii) selecting informative features from a comprehensive list of regions of interest (RoIs). The training of Artemis consists of three stages, with the first two stages being similar to LLaVA. For the final stage, this paper introduces the VideoRef45K benchmark, comprising 45,000 video question-answer pairs, with box-level prompts and answers for complex videos. Experiments demonstrate the promising performance of Artemis across various quantitative metrics, including BERT score, BLEU, and more.

**Strengths:**

This paper is well-written and easy to follow. The authors present a straightforward solution to address video-based referential understanding, a relatively unexplored research area so far as shown by the authors. The method adopted is simple yet aligns well with the paper's motivation. Additionally, the experimental section is robust and well-executed.

**Weaknesses:**

1. As far as I know, Multi-Object Tracking (MOT) is far from satisfying in accurately tracking target regions, particularly in challenging scenarios such as motion blur and occlusion. Although this paper mentions these issues in its limitations section, it does not discuss them in detail. My concern is that the method performs well on the presented benchmarks because the scenarios to be tracked are relatively simple. It would be beneficial if the authors could provide the performance of HQTrack on these benchmarks or offer examples showing the method's efficacy in complex scenarios (e.g., multi-person tracking).
2. This paper does not delve deeply into learning the temporal relationships among tracked Regions of Interest (RoIs). Therefore, the temporal encoding knowledge of these RoIs mainly derives from HQTrack. Since the improvement of this model in video-based referential understanding primarily stems from its better comprehension of temporal dynamics, it raises the concern that the benefit may come from HQTrack rather than the model itself.
3. Comparisons with other models (e.g., Shikra, Merlin) seem somewhat unfair, as these models do not utilize the dynamic knowledge from HQTrack. A fairer comparison setting would enhance the validity of the results.

I am willing to upgrade my rating, if the authors can address my above concerns.

**Questions:**

I am a bit surprised that MLLMs for video referring and grounding have not been explored yet. As I am not very familiar with referential understanding, I look forward to more feedback from other reviewers. If this paper does not overlook any important references, I am willing to upgrade my rating.

**Limitations:**

Please refer to weakness.

---

> ### Author Rebuttal · Authors · 2024-08-07
>
> Thank you for recognizing the contributions of our work. We deeply appreciate your constructive comments and have provided point-to-point responses below. We hope our responses address all your concerns, and further comments are welcomed.
>
> **Q1:** *As far as I know, Multi-Object Tracking (MOT) is far from satisfying in accurately tracking target regions, particularly in challenging scenarios such as motion blur and occlusion. Although this paper mentions these issues in its limitations section, it does not discuss them in detail. My concern is that the method performs well on the presented benchmarks because the scenarios to be tracked are relatively simple. It would be beneficial if the authors could provide the performance of HQTrack on these benchmarks or offer examples showing the method's efficacy in complex scenarios (e.g., multi-person tracking).*
>
>
> **A1:** Good suggestion! Indeed, none of the existing MOT algorithms are close to satisfactory. The original tracking results in HC-STVG were produced by SiamRPN, an early video tracking model with lower accuracy. Although we upgraded the tracking algorithm to HQTrack, it may still fail to track all the objects, especially in complex scenarios.
>
> During the rebuttal, we delved into the test set of HC-STVG and found several examples of multi-person tracking in complex scenarios. We show the tracking and video-based referring results in Figure 19 in the `attachment`. As shown, HQTrack sometimes fails to track the object throughout the video clip -- due to the unavailability of ground-truth labels, we cannot quantitatively compute its accuracy. Regarding the referring results, Artemis produces correct descriptions as long as the tracked boxes are accurate. Interestingly, even when the object is missing in some frames, Artemis can (sometimes) produce correct descriptions based on the visual information from other frames.
>
>
> We will add these examples and analysis to the final paper.
>
>
> **Q2:** *This paper does not delve deeply into learning the temporal relationships among tracked Regions of Interest (RoIs). Therefore, the temporal encoding knowledge of these RoIs mainly derives from HQTrack. Since the improvement of this model in video-based referential understanding primarily stems from its better comprehension of temporal dynamics, it raises the concern that the benefit may come from HQTrack rather than the model itself.*
>
>
> **A2:** Thanks for the question. We totally agree that incorporating richer temporal information and/or knowledge is beneficial for video understanding.
>
> (1) Indeed, our work did not introduce an extra module to formulate the temporal information of the tracked RoI, *e.g.* how it changes throughout the video clip. During the rebuttal, we investigated more examples and found that the model has acquired a preliminary ability to describe the temporal patterns of a video -- see Figure 16 in the `attachment`, where Artemis produces reversed descriptions (a woman walking *down* and *up* the stairs) when the input video is played through the regular and reversed directions. This implies that the MLLM can learn extra temporal knowledge beyond the tracked RoIs. Such abilities may stem from the self-attention module of the MLLM that summarizes the sequential visual features.
>
> (2) An important discovery of our work is that making  proper use of temporal knowledge can largely boost the accuracy of video-based referring. We show a preliminary solution (*i.e.* using an off-the-shelf tracking algorithm to compensate the temporal knowledge), and our efforts reveal a future direction to equip the MLLMs with this ability (*e.g.* one can prompt MLLMs to track the referred objects). Respectfully, we believe that introducing a tracking algorithm (*e.g.* HQTrack) and making the system work is part of our technical contribution.
>
> We will explore stronger solutions in the future. The above discussions will be added to the final paper.
>
>
> **Q3:** *Comparisons with other models (e.g., Shikra, Merlin) seem somewhat unfair, as these models do not utilize the dynamic knowledge from HQTrack. A fairer comparison setting would enhance the validity of the results.*
>
>
> **A3:** Good question! Actually, in evaluating Shikra and Merlin for comparison, we used the same set of tracked bounding boxes (produced by HQTrack) to compute the required visual information: for Merlin, visual features of the entire image were fed into the MLLM together with the bounding box information (in texts); for Shikra which cannot process multiple images simultaneously, both the images and tracked bounding boxes were provided. Therefore, both Shikra and Merlin made use of the dynamic knowledge from HQTrack and thus the comparison was fair to our best efforts.
>
> We will add the above clarification to the final paper.
>
>
> **Q4:** *I am a bit surprised that MLLMs for video referring and grounding have not been explored yet. As I am not very familiar with referential understanding, I look forward to more feedback from other reviewers. If this paper does not overlook any important references, I am willing to upgrade my rating.*
>
>
> **A4:** Here we offer some information for your reference. To the best of our knowledge, two related papers exist prior to our work, namely, PG-Video-LLaVA and Merlin, both of which have been cited in the original submission (see Lines 78--81 in Section 2). PG-Video-LLaVA used off-the-shelf detectors to perform grounding, but the model itself did not have the ability to perform fine-grained video understanding. Merlin studied video-based referring, but it required three manually specified input frames, incurring extra burden for users. Additionally, we also validate the advantage of Artemis over Merlin, the direct competitor.

---

> > ### Comment · Reviewer_CDQu · 2024-08-12
> >
> > The authors' feedback has addressed my concern, I choose to raise my rating.

---

> > > ### Author Response · Authors · 2024-08-12
> > > **Thanks**
> > >
> > > We are happy that our response addressed your question. We appreciate your support for our work.

---

### Official Review · Reviewer_U8HM · 2024-07-17

**Soundness:** 3
**Presentation:** 4
**Contribution:** 4
**Rating:** 7
**Confidence:** 4

**Summary:**

The paper proposes to bring fine-grained understanding to multimodal LLMs (MLLMs) by introducing video-based referential understanding task. The paper motivates with the drawbacks of current image- and video-based MLLMs, and the need for region-specific features to answer region-specific questions . The proposed approach expands the set of video-level features with target region-specific features (box-level prompts) via tracking (HoITrack), alignment (RoIAlign) and selection (clustering). The paper ablates the effectiveness of tracking and selection criteria for improved performance.

**Strengths:**

- The paper is well-written, clear, and the motivation for target-specific features is well presented

**Weaknesses:**

- A fair baseline
    - It’s great that the paper provides comparison with image-based MLLMs by extending them to videos and using an LLM to obtain video-level answers
    - As for the multi-frame approach, it appears that MERLIN is the closest baseline approach. And there seems to be some intersection of pre-training data used for Artemis and MERLIN that includes GOT10K, LaSOT, MeViS
    - But the evaluation of the video-based referring ability is done on the test set of HC-STVG (lines 197-198), and the train set of HC-STVG is included in the pre-training data for Artemis but not for MERLIN
    - Since MERLIN was not trained or fine-tuned on HC-STVG, it does not seem to be a fair, apples-to-apples comparison that MERLIN be evaluated on the test set of HC-STVG
    - On top of that, datasets have biases which includes different label spaces, scene and setting (e.g. HC-STVG is collected from movies), annotation gathering (which can also result in different caption distribution, and hence biasing the eval metrics)
- Ablation
    - Assuming “w/o” in Table 2 means that there is no RoI selection (not mentioned or defined in the text), which I’m assuming to mean that there is no <track-instruction>. If the above assumption is true, “w/o” sets up a strong baseline (even better than MERLIN)
    - In any case, it seems that a careful ablation study is missing
        - (1) w/o <track-instruction>
        - (2) w/ <track-instruction> but where <region> features are not RoI features but the key-frame features. This is to establish whether the improvement is brought forth by key-frame selection or region-level features specifically
- Visualization and sanity check post-training
    - Do the authors have an example of a video with multiple different regions? Mainly this is to inspect how the model response changes with selection of different regions in the same video, and whether it doesn’t degenerates to the same response?
    - Less important, but do the authors have an impression on if the video is reversed, does the caption change?

- Human evaluation
    - Lastly, to substantiate the effectiveness of the approach, did the authors think about a human evaluation study on accuracy / relevance of predicted captions to the video-question pair?
        - This could also be done within your group and with anonymized predictions (meaning the human evaluator doesn’t know what prediction is from what model)


## Minor
- A bit more about the architectural details would have been great (at least in supplemental), especially the tokenization process and whether / how start-end tokens for <instruction> were used
- Do the authors have some statistics on the object category of <region> in the test set of HC-STVG?
    - This is mainly to identify what biases the models are dealing with, and whether those biases skew heavily in one direction. For example, "person" may be the majority category
    - Similarly, any statistics of actions being performed and asked in questions?
- Lastly, do the authors have a quantitative breakdown of performance to understand where the model fails?

**Questions:**

See Weaknesses

**Limitations:**

Yes

---

> ### Author Rebuttal · Authors · 2024-08-07
>
> Thank you for recognizing the contributions of our work. We deeply appreciate your constructive comments and have provided point-to-point responses below. Further comments are welcomed.
>
> **Q1:** *A fair baseline.*
>
> **A1**: During the rebuttal, we fine-tuned Merlin on the same data (*i.e.* VideoRef45K) using LoRA (same as Artemis). We extracted 5 key frames with a bounding box for each clip, aligning with Artemis which uses 5 RoI tokens.
>
> Results are shown in the following table. Fine-tuning brings a significant improvement to Merlin, but the metrics are still lower than Artemis. The reason is two-fold. (1) Artemis introduces RoI tracking, clustering, and selection to obtain accurate localization, so that extracted visual features are of higher quality. (2) Merlin only sees 3--8 frames, while Artemis' encoding method (see Lines 116--122) preserves richer information.
>
> |Method|BLEU@4|METEOR|ROUGE_L|CIDEr|SPICE|
> |:---:|:---:|:---:|:---:|:---:|:---:|
> |Merlin|3.3|11.3|26.0|10.5|20.1|
> |Merlin (ft)|9.7|14.2|35.7|35.1|21.9|
> |Artemis (ours)|15.5|18.0|40.8|53.2|25.4|
>
> We will add them to the paper.
>
> **Q2:** *Ablation.*
>
> **A2**: There are misunderstandings. Explanations below (we will update the notations).
>
> First of all, "w/o" is **not** the baseline, but indicates the option that utilizes the RoI features to encode the referred object based on Video-ChatGPT. That said, "w/o" is a part of the proposed method, which reports higher performance than Merlin.
>
> To facilitate a more intuitive comparison, we added a new baseline. Given an object of interest, we enclosed its location in each frame with a rendered red rectangle, encoded the video using Video-ChatGPT, and asked "What is the object in the red rectangle doing in this video?". As shown, this "baseline" achieves slightly lower results than that of "w/o".
>
> We also added a new option named "w/ <track-instruction>", where the <region> features were replaced with visual features (the `CLS` token of CLIP-ViT-L/14) in the key frames. As shown, there is a performance drop compared to Artemis. This is because the RoI features are of higher quality unlike the whole-frame features impacted by background.
>
> |Method|BLEU@4|METEOR|ROUGE_L|CIDEr|SPICE|
> |:---:|:---:|:---:|:---:|:---:|:---:|
> |baseline|11.2|16.3|34.9|23.8|21.4|
> |w/o|13.9|16.9|39.1|43.7|23.2|
> |w/ \<track-instruction\>|13.9|16.9|38.2|42.1|23.1|
> |Uniformly|14.2|17.2|39.4|44.5|23.6|
> |Artemis (Ours)|15.5|18.0|40.8|53.2|25.4|
>
> **Q3:** *Visualization and sanity check.*
>
> **A3**: Figure 15&19 (`attachment`) shows how Artemis produces different (and correct) answers for different <region>s, as long as tracking is correct. Figure 16 shows how Artemis produces correct descriptions for regular and reversed videos, *i.e.* the woman is walking *down* the stairs in the regular video, and walking *up* the stairs in the reversed video. We will add them to the paper.
>
> **Q4:** *Human evaluation.*
>
> **A4**: We randomly selected 100 videos from the HC-STVG test set. For each case, we applied Artemis and the fine-tuned Merlin (see **A1**) to produce the answers, and asked a person to evaluate their quality (video and ground-truth are provided). The evaluator is **not** aware of which answer is from which model. The evaluator gives a score of 1--5 to each answer (1=worst and 5=best). On average, Artemis and fine-tuned Merlin score 3.36 and 2.65, respectively. Artemis wins Merlin in 53/100 cases, and loses in 21/100 cases. We will add them to the paper.
>
> **Q5:** (Minor) *More architectural details.*
>
> **A5**: We will add them to the paper.
>
> Artemis consists of three components: a visual encoder (CLIP-ViT-L/14), a large language model (Vicuna-v1.5-7b), and a RoI feature extraction module. The RoI feature extraction module utilizes the visual features of 4 layers of the visual encoder to extract RoI features and passes them to a linear layer to obtain the RoI token.
>
>
> For text tokenization, we use Vicuna-v1.5's built-in tokenizer. To insert the video and RoI features into Q&A, we use \<image\> as a placeholder for video features, with its ID=-200 indicating the position of video features. Similarly, we use <bbox> for ROI tokens with ID=-500. No additional tokens like <image-start> and <image-end> are used.
>
> **Q6:** (Minor) *Statistics on object category/actions.*
>
> **A6:** All referred objects in the HC-STVG test set are humans (referred to as different nouns like man/woman). In the training set (VideoRef45K), there are other categories (*e.g.* animals/vehicles) -- see Figure 8 in Appendix A; the actions of these objects are much simpler compared to humans in HC-STVG, so we chose HC-STVG test set to challenge Artemis and others.
>
> We used SpaCy to extract and count the action types in the HC-STVG test set. The distribution of 384 actions is shown in Figure 17 (`attachment`).
>
> **Q7:** (Minor) *Breakdown of failure.*
>
> **A7:** We defined five error types:
> * Temporal perception error, *e.g.* output is "going down", ground-truth is "going up".
> * Incomplete action error, *e.g.* only mentioning "walking" but omitting "turning" in the ground-truth.
> * Action recognition error, *e.g.* output is "running", ground-truth is "jumping".
> * Object recognition error, *e.g.* output is "woman", ground-truth is "man".
> * Multi-object interaction error, *e.g.* output is "woman gave sth. to man", ground-truth is "man gave sth. to woman".
>
> We provided the output and ground-truth of each failure case to GPT-3.5 to get the error type. Statistics are shown in Figure 18 (`attachment`). Most frequent failures come from incomplete action and object recognition, albeit RoI tracking has alleviated them to some extent. Besides, object recognition errors mainly happen in the interacting object, *e.g.*, "the man touches the woman's face" is misrecognized as "the man touches the man's face". This indicates limited improvement of RoI tracking on interacting objects. This enlightens a future direction.
>
> We will add them to the paper.

---

### Author Rebuttal · Authors · 2024-08-07

We thank reviewers for their meticulous work and the insightful comments provided to us.

All reviewers acknowledged the novelty and contributions of the proposed approach (Artemis).

**Reviewer U8HM&CDQu:** The paper is **well-written and clear** and the **motivation is well presented**.

**Reviewer CDQu:** The **method is simple yet aligns well with the paper's motivation**. The **experimental section is robust and well-executed**.

**Reviewer xzsp:**  A video-based referential understanding dataset, VideoRef45K, is established. **It facilitates the development of the area**, providing referential pretraining data.  The design of target-specific feature branch in the model architecture is **well-motivated**. Combined with existing video-language models, Artemis can perform multi-round video understanding with grounding and long-form video understanding.

The major concerns and suggestions lie in the **fair comparison with other models** (**U8HM**, **CDQu**), **more ablative studies** (**U8HM**), **more model comparisons** (**xzsp**), the **influence of the tracking model** in Artemis (**CDQu**), and more examples and **implementation details** (**U8HM**, **CDQu**).

During the rebuttal, we have carefully considered reviewers' every feedback and provided more experiments and ablation studies, as suggested. We believe these point-to-point responses can address reviewers' concerns and further enhance our work. We also include a PDF file (referred to as the `attachment`) with additional experimental results to support our responses.

---

### Decision · Program_Chairs · 2024-09-25

**Decision:**

Accept (poster)

**Comment:**

The paper originally received borderline scores leaning towards acceptance. The authors provided a detailed rebuttal and the paper has been extensively discussed by the reviewers. Reviewers CDQu and xzsp raised some concerns about comparisons to the previous works and missing baselines (in particular for what concerns the video-language models) and the contributions of the tracked ROIs. The authors successfully addressed the issues and both the reviewers where satisfied by the rebuttal and increased their scores to 7:Accept. The AC carefully revised all the material and there has been also some further discussion with the reviewers. Overall this is a solid submission and the authors are encouraged to follow and implement all the reviewers' suggestion in the final revision of the paper.